# Structure Investigation of La, Y, and Nd Complexes in Solvent Extraction Process with Liquid Phosphine Oxide, Phosphinic Acid, and Amine Extractants

**Amilton Barbosa Botelho Junior \*** , **Natália Olim Martins da Silva, Jorge Alberto Soares Tenório and Denise Crocce Romano Espinosa**

Department of Chemical Engineering, Polytechnic School, University of Sao Paulo, Sao Paulo 05508-080, Brazil; natalia.olim@usp.br (N.O.M.d.S.); jtenorio@usp.br (J.A.S.T.); espinosa@usp.br (D.C.R.E.)
\* Correspondence: amilton.junior@usp.br

**Abstract:** The main challenge in separating REEs through hydrometallurgical processes is their chemical similarities. Despite the literature widely presenting the possibilities for organic extractants, there is a lack of evaluation of the structures formed between the REEs and the extractants. The present study aimed to evaluate different extractants (neutral, anionic, and acid extractants) for separating La, Y, and Nd. The extraction efficiencies were evaluated, and the structure investigation was carried out in FT-IR. From the results obtained, it is clear that the extraction order is Alamine 336 <<< Cyanex 272 < Cyanex 923, where both acid extractants were more selective for Y than for La and Nd. The extraction achieved 99% at pH 5.0 in nitric acid media, and a Y/La ratio of 2 and a Y/Nd ratio of 4 using Cyanex 923. The present study also elucidated the organometallic complexation between Cyanex 923 and Cyanex 272 with Y and La, which may improve separation processes to obtain critical metals from primary and secondary sources.

**Keywords:** Cyanex 923; Cyanex 272; Alamine 336; critical metals; REE





## 1. Introduction

The rare earth elements (REE) are a group of elements, including Sc, Y, and lanthanides, divided into light rare earth elements (LREE)—from La to Eu, and heavy rare earth elements (HREE)—from Ga to Lu, including Y [1,2]. Among these elements, some of them have crucial importance to society. For instance, Y is used in the lighting market for fluorescent bulbs [3] as well as LED/OLED gadgets [4,5]; La is also used in lamp materials with Y, as well as batteries and in the refractive industries [6,7]; and Nd is the main element with Fe and B for permanent magnets [8]. Due to their demand and low substitution rates, the leading economies worldwide consider these elements as critical/strategic, including the European Union [9], USA [10], Brazil [11], and Australia [12].

Due to the economic importance and risk of interruption in the supply chain associated with these elements, which are important for net-zero technologies, the search for new sources and the development of new processes have received attention [4,13,14]. Hydrometallurgical processing is the main route to obtain REEs, as these elements are usually present in low concentrations in both primary (minerals and ores) and secondary sources (mining residues and e-wastes). In the case of primary sources or mining residues such as minerals/ores, the material is first converted into an acid-soluble extract, which may be acid or alkali, and then undergoes chlorination or oxidative roasting [1]. In the case of e-wastes, physical treatments are necessary before hydrometallurgical treatment [15].

After that, the materials follow a similar flowchart: leaching and ion exchange separation. The leaching reaction usually occurs in acid media due to the thermodynamic conditions; these include mineral acids ($H_2SO_4$, $HNO_3$, and HCl), salts (NaCl and $(NH_4)_2SO_4$), and organic acids (citric and malic acids) [14,16]. Furthermore, solvent extraction is the

main technique used to achieve REE separation. Although ion exchange resins have been explored [13], solvent extraction is the dominant technology to separate the elements, despite their chemical similarities. It occurs by contact between the aqueous phase containing REEs and the organic phase containing the extractant [17].

Table 1 presents the state-of-the-art solvent extraction separation for La, Y, and Nd. The most common extractants used are D2EHPA, Cyanex 272, and TOPO, as the organophosphate extractants are selective for REEs [1]. Innocenzi et al. (2017) used D2EHPA and Cyanex 272 in kerosene for the selective separation of Y from a fluorescent lamp leach solution, in which D2EHPA reached the best results, achieving a 100% extraction rate after three contacts [18]. Batchu and Binnemans (2018) evaluated Dy separation using D2EHPA, which was demonstrated to be more selective than Nd [19], while Sun et al. (2018) demonstrated that Cyanex 272 may be used for Nd extraction instead of Dy [20].

**Table 1.** A literature review of solvent extraction of La, Y, and Nd.

| Ref. | Element | Extractant | Experimental Parameters | Extraction Rate |
|------|---------|------------|-------------------------|-----------------|
| [21] | La | D2EHPA (bis(2-ethylhexyl)phosphoric acid) in NB18C6 (cyclohexane using 2-nitrobenzo-18-crown-6) | pH 3.0, 0.01 mol/L D2EHPA, 25 °C, A/O 1/1 | 100% |
| [22] | | Cyanex 272 (bis(2,4,4-trimethylpentyl)phosphinic acid) with TOPO (trioctylphospine oxide) in toluene | pH 5.6, 0.0215 mol/L Cyanex 272, 29 °C, A/O 1/1 | not informed |
| [23] | | D2EHPA (bis(2-ethylhexyl)phosphoric acid) in n-heptane | pH 4.0, 20% D2EHPA, 25 °C, A/O 1/1 | 100% |
| [18] | Y | D2EHPA (bis(2-ethylhexyl)phosphoric acid) and Cyanex 272 (bis(2,4,4-trimethylpentyl)phosphinic acid) in kerosene | pH 0.02, 20% D2EHPA, 25 °C, A/O 1/1 | 100% * |
| [24] | | Aliquot 336 (N-Methyl-N,N,N-tri-octylammonium chloride) with thiocyanate or nitrate media in kerosene | pH 3.0, 20% Aliquat336 from thiocyanate and nitrate medium, 25 °C O/A = 1 | 95% |
| [25] | | HPOAc (2-(bis((2-ethylhexyl)oxy)phosphoryl)-2-hydroxy- acetic acid) and P507 (2- Ethylhexyl phosphoric acid mono-2-ethylhexylester) in kerosene | pH 5.0, 0.5 mol/L HPOAc, 25 °C O/A = 1 | 60% |
| [26] | | D2EHPA (bis(2-ethylhexyl)phosphoric acid) in kerosene | 1 mol/L $HNO_3$, 0.3026 mol/L D2EHPA, 25 °C, A/O 1/1 | 93% |
| [19] | Nd | D2EHPA (bis(2-ethylhexyl)phosphoric acid) evaluating diluents | 1 mol/L NaCl, 1 mol/L D2EHPA in the aliphatic diluent, 20 °C, A/O 1/1 | 29% ** |
| [20] | | Cyanex 272 (bis(2,4,4-trimethylpentyl)phosphinic acid) in kerosene | pH 3.0, 0.5 mol/L Cyanex 272 in kerosene, 25 °C, A/O 1/1 | 7.4% *** |
| [27] | | TOPO (trioctylphosphine oxide) in kerosene | 500 mol/m$^3$; $HNO_3$, 500 mol/m$^3$; TOPO in kerosene, 25 °C, A/O 1/1 | 100% |

* after three-stage cross current extraction, ** evaluating diluents on extraction, *** Dy extraction.

Innocenzi et al. (2017) used D2EHPA and Cyanex 272 in kerosene for the selective separation of Y from a fluorescent lamp leach solution, where in which D2EHPA reached the best results, achieving a 100% of extraction rate after 3 three contacts [18]. Batchu & and Binnemans (2018) evaluated Dy separation using D2EHPA, which was demonstrated to be more selective than Nd [19], while Sun et al. (2018) demonstrated that Cyanex 272 may be used for Nd extraction instead of Dy [20].

Among the phosphate extractants, Cyanex 923 (a mixture of phosphine oxides) has been demonstrated as a potential extractant with a high extraction capacity for REEs, low water solubility, and good solubility in hydrocarbons [28]. For this reason, it has been evaluated for the extraction of REEs in different processes [29–31].

The literature reports several studies regarding the separation and purification of Y, La, and Nd by solvent extraction. However, more is needed to evaluate the chemical structures of these elements and the organic extractants–organometallic complexation mechanism. Santanilla et al. (2021) investigated the structures formed between Ni and Co complexes with Cyanex 272 and Versatic 10. Their investigation by extraction efficiencies and FT-IR analyses demonstrated that hydrated complexes are formed between the metallic ions and the extractants [32].

Although there is a wide range of literature about using these organic extractants to extract rare earth elements, none of them has addressed the selectivity of REE extraction. They elucidated the organometallic complexation between organic extractants and REEs. This highlights the novelty of the present study concerning industrial applications to obtain rare earth elements from primary and secondary sources, which include mining wastes and e-wastes.

## 2. Materials and Methods

### 2.1. Materials

Three organic extractants were used in this study without purification (purity $\geq$ 95%): Cyanex 923 (mixture of four trialkylphosphine acids—TRPO), Cyanex 272 (bis(2,4,4-trimethylpentyl)phosphinic acid), and Alamine 336 (tertiary amine). The physicochemical properties of the extractants and chemical structures are reported in the literature [31]. For the experiments, synthetic solutions were prepared using oxides of lanthanum ($La_2O_3$), neodymium ($Nd_2O_5$), and yttrium ($Y_2O_3$) dissolved in $HNO_3$ or $HNO_3$-HCl diluted in ultra-pure water. The pH was adjusted using $HNO_3$ or NaOH (in beads). The stock solutions (500 mg/L) were prepared before the experiments. The organic phase was prepared with a known volume of organic extractant and kerosene (Casa Americana). The samples from the solvent extraction experiments were analyzed in energy dispersive X-ray fluorescence (EDX-7200 Shimadzu), and infrared spectroscopy analysis was carried out from 4000 cm$^{-1}$ to 400 cm$^{-1}$ (Bruker Tensor 27), coupled with attenuated total reflectance (ATR). The pH was measured using an Ag/AgCl electrode in 3 mol/L (Sensoglass).

### 2.2. Methods

The thermodynamic analysis was carried out using Hydra-Medusa software in the conditions at which the experiments were conducted. The goal was to analyze the species of REEs that are present in the solution. Although the thermodynamic data for these elements are less complete than most common elements (such as Fe, Ni, and Co), such analysis is important to evaluate the cationic and anionic species present in the solution [3,33].

The experiments were carried out in batches by mixing 15 mL of synthetic solution and previously prepared 15 mL of organic phase (aqueous/organic ratio: 1/1). After 15 min under magnetic stirring with the pH controlled at the desired value, the mixture was placed in a separatory funnel for 10 min. Then, the aqueous phase was filtered using filter paper for organic retention (Whatman 3).

The effect of pH was evaluated from 1.0 to 5.0 using an organic extractant diluted in kerosene (10%). Selective extraction between REEs was also evaluated in binary solutions by varying the ion concentration. The extraction of La, Nd, and Y was evaluated for three different types of extractants: dialkyl phosphinic acid (Cyanex 272), liquid phosphine oxide (Cyanex 923), and amine-based extractants (Alamine 336), and their chemical structures are presented in Figure 1. After the phase separation, the aqueous sample was analyzed by EDX, and the organic phase was analyzed by FT-IR.

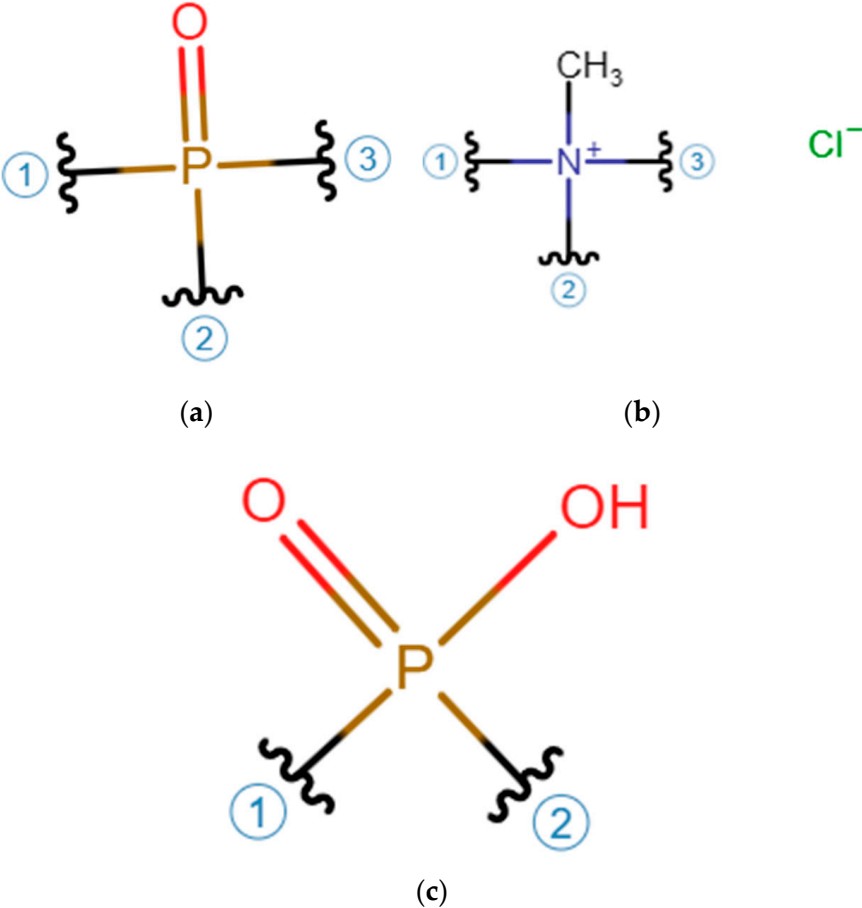

**Figure 1.** Chemical structures of (**a**) Cyanex 923, where (1), (2), and (3) are hexyl or octyl, (**b**) Alamine 336, and (**c**) Cyanex 272 [31].

The amount of ions extracted was calculated by mass balance, as depicted in Equation (1), where $C_i$ and $C_f$ are the concentration of the REEs before and after the extraction experiment, and $V_i$ and $V_f$ are the volume of the aqueous phase before and after the extraction experiment, respectively. The distribution ratio ($D$) and separation factor ($\beta$) were calculated as depicted in Equations (2) and (3), respectively, where $M_t$ and $M_a$ are the initial and final concentrations of the metallic ions in the aqueous phase.

$$E(\%) = \frac{C_i \cdot V_i - C_f \cdot V_f}{C_i \cdot V_i} \cdot 100\% \tag{1}$$

$$D = \frac{M_t - M_a}{M_a} \tag{2}$$

$$\beta_{1,2} = \frac{D_1}{D_2} \tag{3}$$

## 3. Results and Discussion

### 3.1. Thermodynamic Analysis

Figure 2 shows the speciation diagram for $La(NO_3)_3$ in solution. The potential redox of the synthetic solution was 0.9 V, and the pH in the thermodynamic simulation was evaluated from 0 to 6. As observed for La, the element is present in solution in different cationic species (Figure 2a). The main species are $La^{+3}$ and $LaNO_3^{2+}$ (Figure 2b). Anionic species such as $NO_2^-$, $NO_3^-$, and $OH^-$ are also present in the solution, which does not affect the separation reaction since there is no presence of La ions.

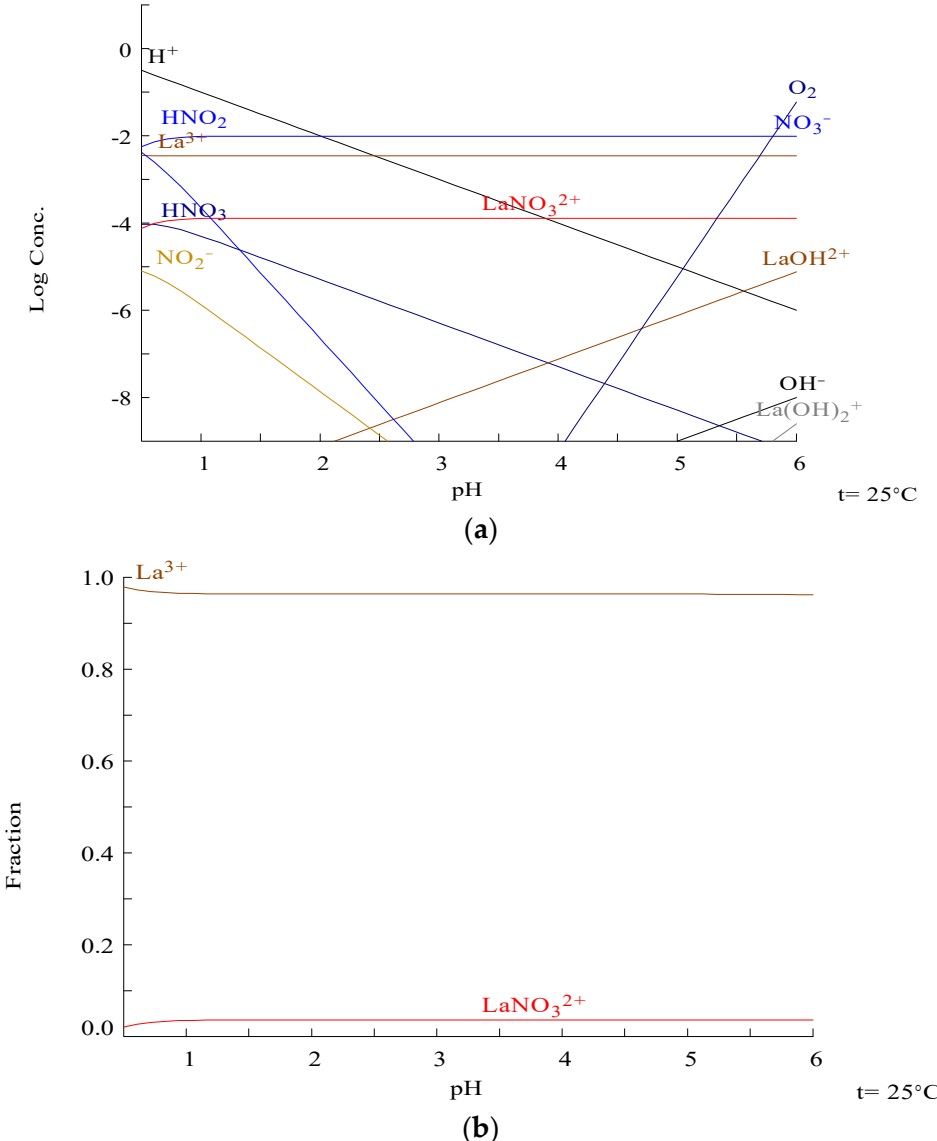

**Figure 2.** Speciation diagram of La(NO$_3$)$_3$ solution for solvent extraction experiments at 25 °C: (**a**) log concentration of the species, and (**b**) fraction of main La species.

In addition, up to pH 6, there was no solid phase, demonstrating that no precipitate was formed during the experiments. As demonstrated by Almeida Neto (2014), the speciation diagram showed that Ni and Cu precipitated in pH 6.5 and 5.5, respectively, which negatively impacts the ion exchange separation [34]. In addition, there are cases where anionic compounds may be formed in an acidic pH. De Oliveira Damarco et al. (2020) showed that Pt anionic compounds, such as PtCl$_4{}^{-2}$, are present in solutions from pH 0 to 10, which could affect the separation process. It should be considered in the choice of organic extractant [35].

Figure 3 presents the speciation diagram for Y(NO$_3$)$_3$ in solution, which presented similar characteristics to the La diagram (Figure 2). However, the concentration of nitrate complexes (as YNO$_3{}^{-2}$) is lower than for La, which is stated in Figure 3b, in which the majority of Y compound is Y$^{+3}$. In both cases, hydroxide complexes with a pH above 5 are cationic compounds. Figure 3a demonstrates that YOH$^{+2}$ starts to form at pH 1, while, for La (Figure 2a), the hydroxide complex, LaOH$^{+2}$, starts to form at pH 2.

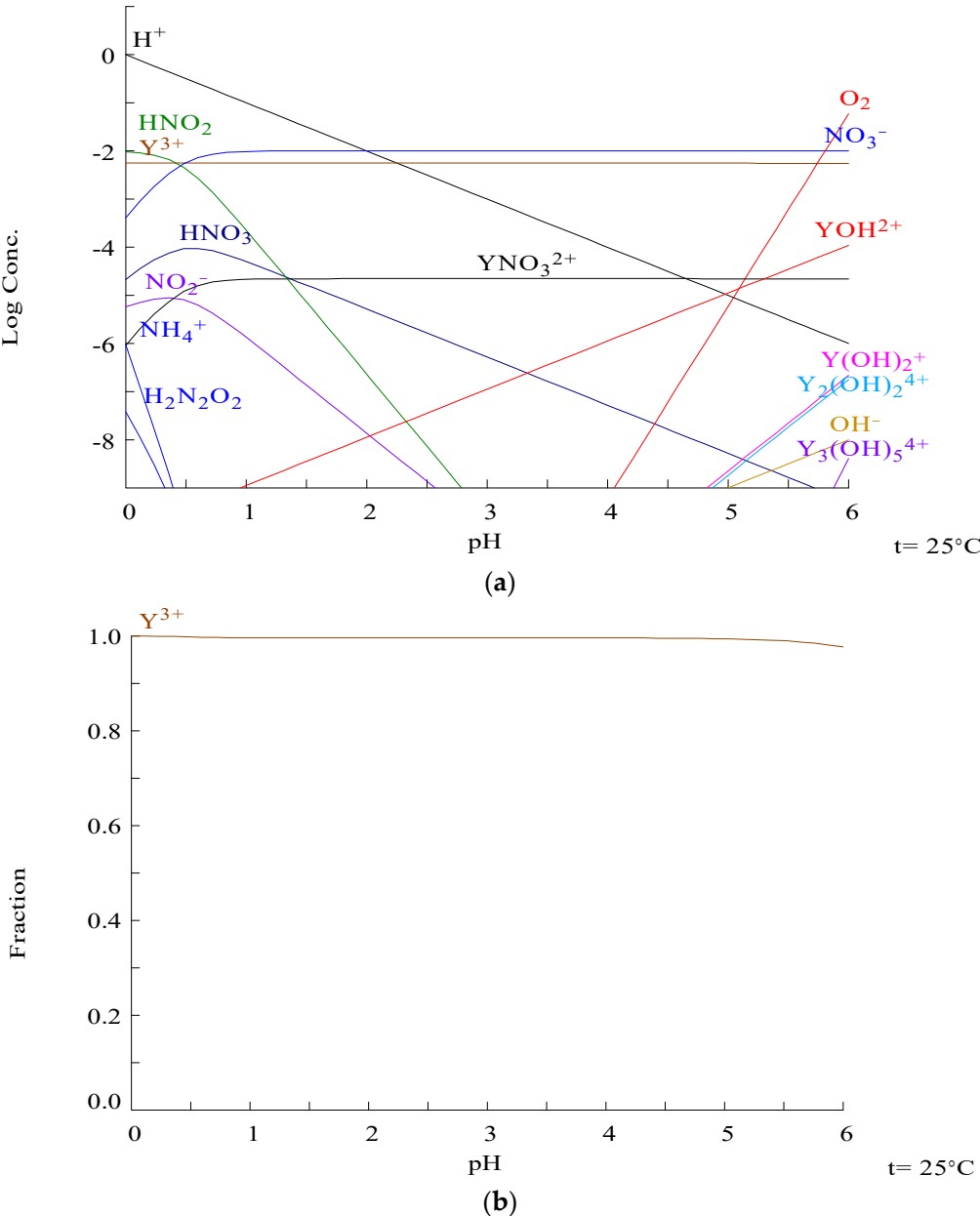

**Figure 3.** Speciation diagram of Y(NO$_3$)$_3$ solution for solvent extraction experiments at 25 °C: (**a**) log concentration of the species, and (**b**) fraction of main Y species.

Figure 4 presents the thermodynamic simulation for Nd(NO$_3$)$_3$ in solution. As observed for La and Y, there is no precipitation from pH 0 to 6, and the main compounds in the solution are cationic. The behavior of Nd is similar to La, as both are considered as LREEs, having chemical similarities. At the same time, Y has been classified as HREE, due to its common chemical and physical affiliations with the heavy REEs [1]. Agarwal et al. (2018) evaluated the solvent extraction technique for Eu separation by solvent extraction using phosphinic acids. As reported by the authors, in hydrochloric media, the main compounds are cationic, and the increase in Cl ions concentration increased the amount of Cl compounds. In 3 mol/L of Cl$^-$, EuCl$_4$$^-$ and EuCl$_4$ are the majority compounds [36]. However, the present study does not evaluate extreme acid concentrations because the increase in H$^+$ ions concentration reduces the extraction efficiency [37].

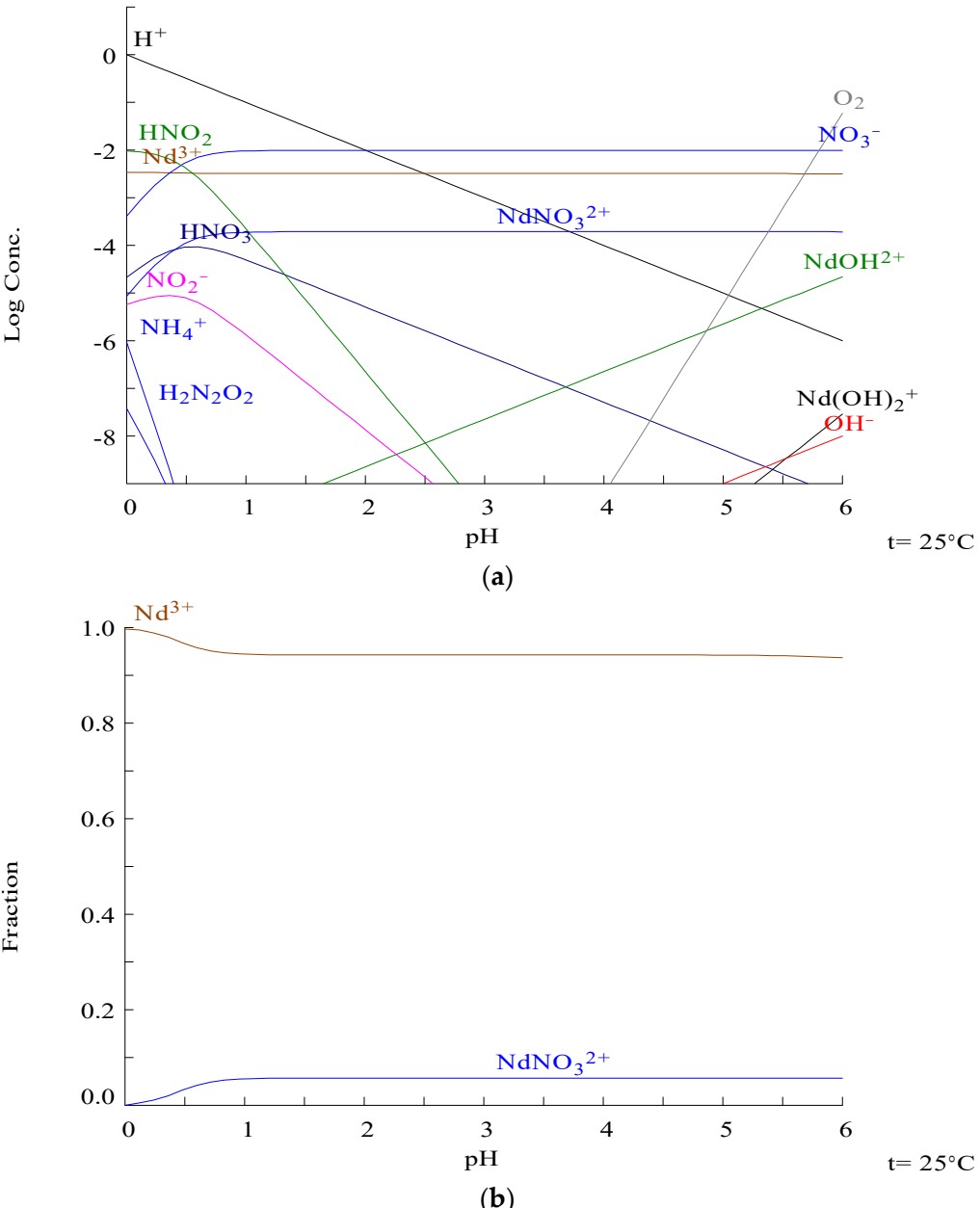

**Figure 4.** Speciation diagram of $Nd(NO_3)_3$ solution for solvent extraction experiments at 25 °C: (**a**) log concentration of the species, and (**b**) fraction of main Nd species.

### 3.2. Extraction of Y, La, and Nd and their Mixtures

The experiments were carried out in batches for 15 min at room temperature (25 °C) and with an aqueous/organic ratio of 1/1. The aqueous phase was filtered and analyzed. The organic extractants tested were Cyanex 923, Alamine 336, and Cyanex 272, and the chemical structures are presented in Figure 1. The effect of pH on metal extraction by Cyanex 923 is presented in Figure 5. Although the extractant is considered neutral by a few authors [38], the extraction of the metals was affected by the pH change when the concentration of cation compounds was in the majority. As the concentration of $H^+$ ions decreases as the pH increases, the competition between them and the metals is lower [31].

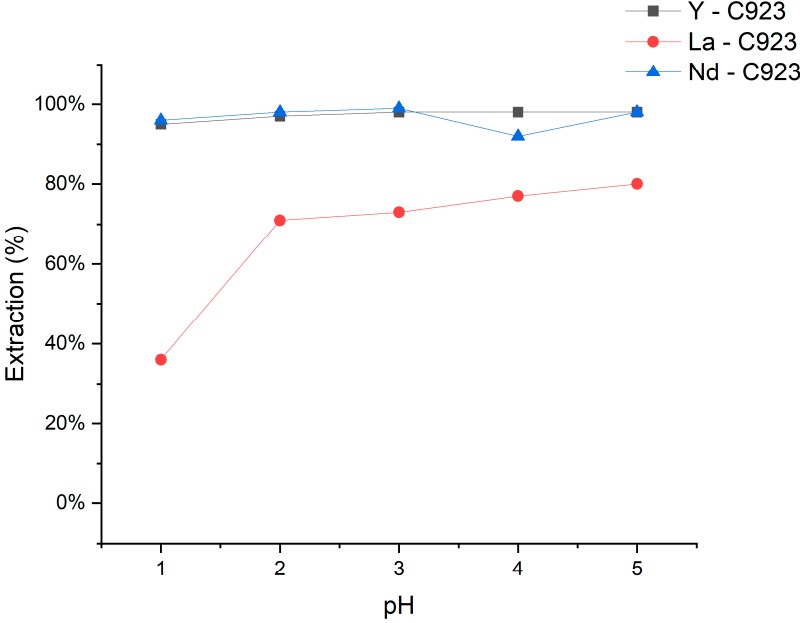

**Figure 5.** Extraction efficiency of Y, La, and Nd by Cyanex 923 (C923) 10%, varying the pH at 25 °C, A/O 1/1 for 15 min.

The extraction of Y and Nd slightly increased, from 95% at pH 1.0 to 98% at pH 5.0. Despite La and Nd having chemical similarities, the extraction of La increased from 36% to 80% as the pH increased from 1.0 to 5.0. As Tunsu et al. (2016) demonstrated, Cyanex 923 is more selective for Y than La—almost 18-fold in the solvent extraction separation of fluorescent lamp leaching [29]. In the presence of Nd and La from a leach solution of bauxite residue, the extraction of Y is more selective using Cyanex 923 [31,39], as depicted in Equation (4) (Cyanex 923 = TRPO) [30,40].

Table 2 depicts the separation factors of the metals. As observed, the Y/La separation factor declined from 2.64 to 1.23 as the pH increased from 1.0 to 5.0, since the extraction of Y remained close to 100% and La lower than 80%. Similar results were obtained for the Nd/La separation factor. The Y/Nd ratio remained almost 1 in all the pH tests.

$$(REE)^{+3}{}_{(aq)} + (NO_3)^{-2}{}_{(aq)} + 3\text{TRPO}_{(org)} \rightarrow (REE)NO_3 \cdot 3\text{TRPO}_{(org)} \tag{4}$$

**Table 2.** Separation factors of Y/La and Y/Nd in monoelementary synthetic solution, varying the pH, using Cyanex 923 at 25 °C, A/O 1/1 for 15 min.

| pH | Y/La | Y/Nd | La/Nd | Nd/La |
|-----|------|------|-------|-------|
| 1.0 | 2.64 | 0.99 | 0.38 | 2.67 |
| 2.0 | 1.37 | 0.99 | 0.72 | 1.38 |
| 3.0 | 1.34 | 0.99 | 0.74 | 1.36 |
| 4.0 | 1.27 | 1.07 | 0.84 | 1.19 |
| 5.0 | 1.23 | 1.00 | 0.82 | 1.23 |

As observed in Figure 5, the extraction of La was influenced by the pH, despite the fact Cyanex 923 is a neutral extractant. This occurs because the nitric acid concentration decreased as the pH increased. As observed in Figure 2, the concentration of $HNO_3$ decreased to zero at a pH around 2.8 and there was a slight increase in nitrate ion formation. From pH 1 to 3, the extraction of La increased from 36% to 70%. As depicted in Equation (4), this results in the equilibrium being pushed to the right for the product formation. Also, as

the pH increases, the H$^+$ in solution competes with the metallic ions for the extractant that was highlighted for the La ions [30].

Figure 6 presents the extraction efficiency of Y, La, and Nd by Alamine 336 (336). As the extractant is anionic, no effect on extraction efficiency was observed as the pH changed from 1.0 to 5.0. La and Nd extraction achieved 52–61% and 22–32%, respectively, while Y extraction slightly decreased from 49% to 28% as the pH increased from 1.0 to 5.0. These elements were probably slightly extracted due to the high concentration of cationic compounds, and the thermodynamic simulations might explain the slightly decreased Y extraction. In Figure 3, at pH 5.0, there are cationic species +4, while the extractant reacts by solvation interaction, as depicted elsewhere [41]. Conversely, Alamine 336 may be used for REE extraction in HCl media, as Cl$^-$ ions complex the metallic ions in solution into anionic compounds [40]. As observed in Figure 6, there is no selectivity in the separation of Y, La, and Nd using Alamine 336.

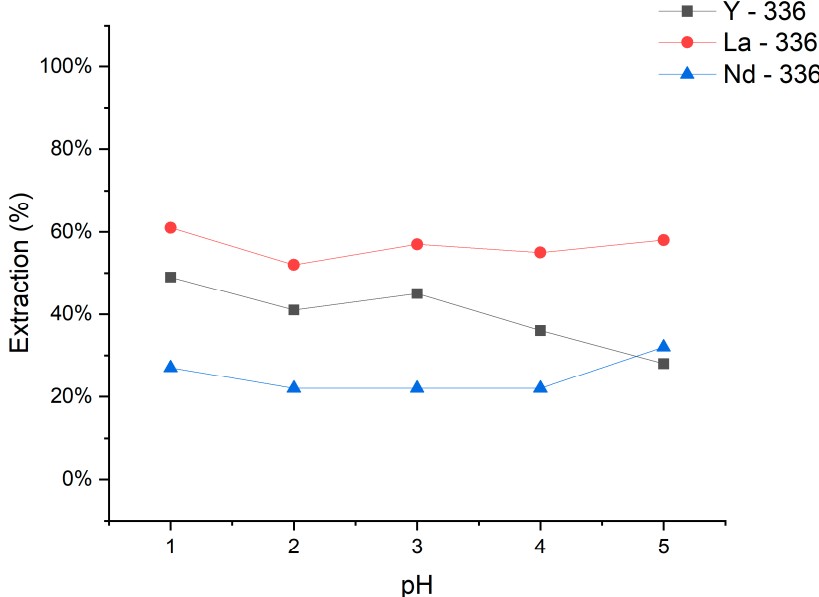

**Figure 6.** Extraction efficiency of Y, La, and Nd by Alamine 336 (336) 10%, varying the pH at 25 °C, A/O 1/1 for 15 min.

Figure 7 depicts the extraction efficiency using Cyanex 272 (C272). As observed, the extractant has more selectivity for Y than for La and Nd at pH 1.0 and 2.0, and then the extraction efficiencies remained similar at close to 100%. Equations (5)–(7) show the extraction of Y, La, and Nd, respectively, by Cyanex 272 [40]. As the reaction occurs, H$^+$ ions are released into the solution. Table 3 presents the separation factor between Y and La, and Y and Nd. It can be seen that Cyanex 272 is more selective for Y than for La and Nd. At pH 1.0, the separation factor was 2.15 for Y/La and 4.02 for Y/Nd, and then declined to 1 as the pH increased. For the La/Nd separation factor, it was achieved at 1.87 at pH 1.0, and, above 2.0, the factor was 1.0. For this reason, in the conditions in which the experiments were carried out, it may be inferred that the order of selectivity is Y > La >> Nd.

$$Y^{+3}{}_{(aq)} + 3(HX)_{2(org)} \rightarrow YX_33HX_{(org)} + 3H^+{}_{(aq)} \tag{5}$$

$$La^{+3}{}_{(aq)} + 3(HX)_{2(org)} \rightarrow YX_33HX_{(org)} + 3H^+{}_{(aq)} \tag{6}$$

$$Nd^{+3}{}_{(aq)} + 3(HX)_{2(org)} \rightarrow YX_33HX_{(org)} + 3H^+{}_{(aq)} \tag{7}$$

**Table 3.** Separation factor of Y/La and Y/Nd in monoelementary synthetic solution, varying the pH using Cyanex 272 at 25 °C, A/O 1/1 for 15 min.

| pH | Y/La | Y/Nd | La/Nd |
|-----|------|------|-------|
| 1.0 | 2.15 | 4.02 | 1.87 |
| 2.0 | 2.04 | 1.85 | 0.91 |
| 3.0 | 1.00 | 1.01 | 1.00 |
| 4.0 | 0.99 | 1.10 | 1.11 |
| 5.0 | 0.99 | 1.01 | 1.02 |

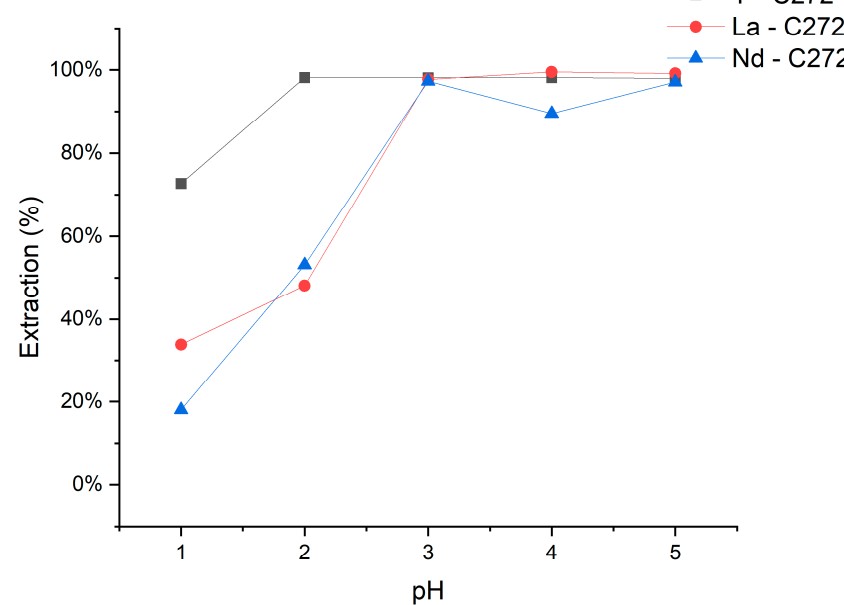

**Figure 7.** Extraction efficiency of Y, La, and Nd by Cyanex 272 (C272) 10%, varying the pH at 25 °C, A/O 1/1 for 15 min.

The main challenge is the Y and La separation, as the separation factor is lower than Y/Nd. In addition, both elements are commonly found together in hydrometallurgical processes, such as in mining or secondary sources (fluorescent and LED lamps) [3,5,42–45]. Nd sources are commonly related to REEs in trace concentrations, such as Pr and Dy [46]. However, the present study demonstrated that both phosphinic extractants may be used for Nd extraction using Cyanex 923 (Figure 5), or Y/Nd separation using Cyanex 272 (Table 3).

The Y/La separation by Cyanex 923 at pH 5.0 was evaluated in an elementary solution at varying concentrations, and the results are presented in Table 4. The extraction of Y was over 96% in all the experiments, while La was zero. The experiments carried out with 100 mg/L and 200 mg/L of La and 500 mg/L of Y reached a high level of La extraction, but, probably due to its low concentration, the extraction process [47] was extremely slow.

Also, comparing the rare earth elements, Cyanex 923 was more selective for Y than for La in all the experiments, even at lower concentrations. Tunsu et al. (2014) demonstrated that Cyanex 923 had a higher selection for Y than other metals in removing leach liquor from fluorescent lamp recycling [30]. According to the authors, the separation factor may be related to the ionic radius of the elements. The ionic radius of Y (180.12 pm) is slightly lower than La (187.91 pm) [48]. In Sc separation from bauxite residue leach solution, Cyanex affects Sc significantly more than Y, and these elements have different ionic radii: 164.06 pm and 180.12, respectively [31]. This indicates that the ionic radius has a significant effect on the separation of these elements by Cyanex 923. On the other hand, Table 2 demonstrates similar extraction factors for Nd and Y, while both have different ionic radii. For this reason, the separation of Y over may be carried out using Cyanex 923 with a high separation factor.

**Table 4.** Separation efficiency of Y/La and Y/Nd in elementary synthetic solution by Cyanex 923.

| Y (mg/L) | La (mg/L) | Y (%) | La (%) |
|---|---|---|---|
| 500 | 100 | 97 | 34 |
| 500 | 200 | 96 | 35 |
| 500 | 300 | 96 | 0 |
| 500 | 400 | 98 | 0 |
| 500 | 500 | 98 | 0 |
| 100 | 500 | 98 | 0 |
| 200 | 500 | 98 | 0 |
| 300 | 500 | 96 | 0 |
| 400 | 500 | 97 | 0 |

Figure 8 presents the species diagram in the elementary Y-La system. As observed, the majority of compounds are $La^{+3}$ and $Y^{+3}$, followed by $YNO_3^{+2}$ and $LaNO_3^{+2}$. Hydroxides were formed in pHs above 5, and all as cationic species. No formation of La-Y complexes was observed, only separated complexes. This indicates that the reaction between the organophosphate extractant and the metallic ions was carried out between the separated metals.

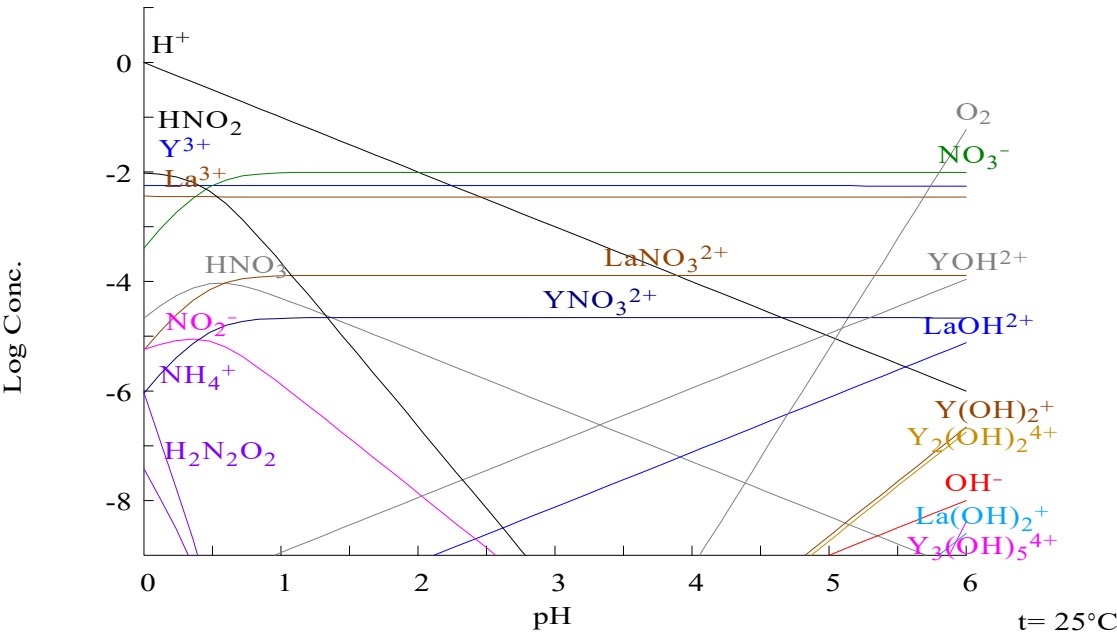

**Figure 8.** Log concentration of the species formed by $Y(NO_3)_3$ and $La(NO_3)_3$ solutions in solvent extraction experiments at 25 °C.

As the concentration did not affect the separation of Y over La ions by Cyanex 923, the separation factor between the ions was evaluated by Cyanex 272. In this study, the effect of pH was evaluated, and the results are presented in Figure 9. The difference among the extractants was observed, as the Y/La separation factor declined from 4 to 2, while, at pH 5.0, the Cyanex 923 extracted all the Y ions selectively. According to the results obtained, the extraction of the rare earth elements follows the order Alamine 336 >>> Cyanex 272 > Cyanex 923. Cyanex 923 was more selective for Y than for La and Nd in the selective separation. Considering industrial processes, La represents the main challenge to obtaining Y, as both are found in spent lamps [5,49] and mining processes [44].

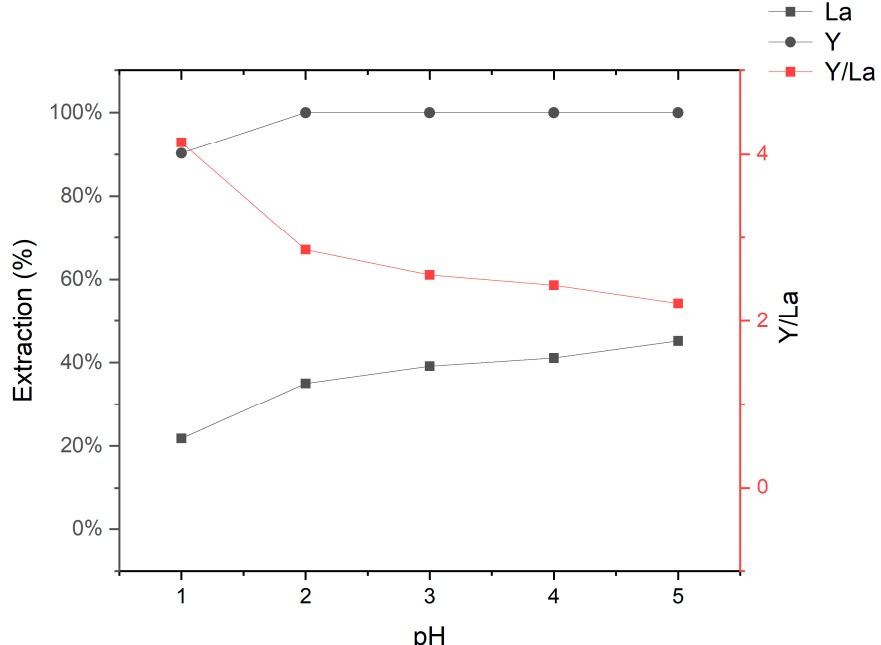

**Figure 9.** Separation efficiency of Y/La and Y/Nd in elementary synthetic solution by Cyanex 272.

### 3.3. FT-IR Analysis

In order to study the difference between the Y and La separation presented in Figure 5, FT-IR analyses were carried out for the organic phases after the extraction process (Figure 10). This identified the following peaks: 2956 cm$^{-1}$ and 2923 cm$^{-1}$, which may be related to the C-H group, 1459 cm$^{-1}$ is related to the C-O-H or CO group, 1378 cm$^{-1}$ is related to the -COO- group, and 1162 cm$^{-1}$ is related to the P=O group [32,50,51]. It was observed that the peak, P=O, was slightly displaced after contact with the rare earth elements.

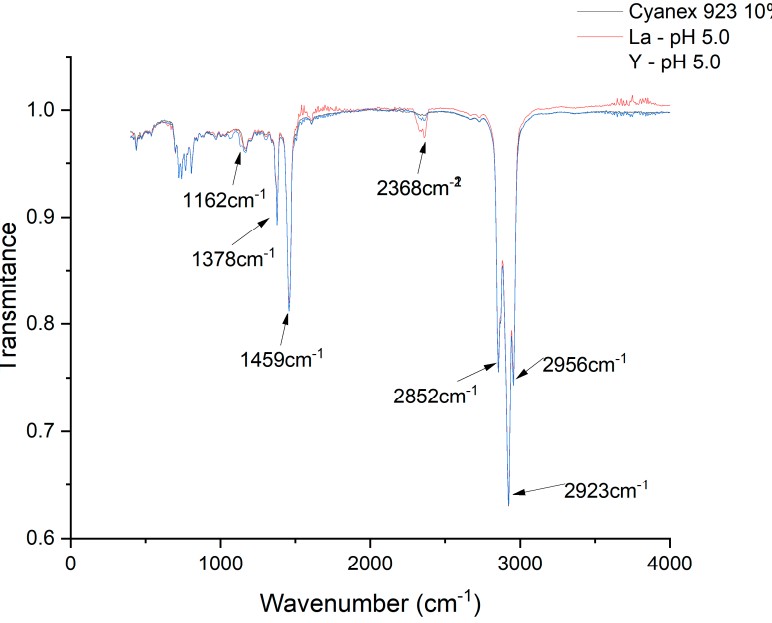

**Figure 10.** FT-IR spectra of Cyanex 923 10% in kerosene, before and after the extraction of La and Y in nitric acid media at pH 5.0.

A few references report a peak at 2368 cm$^{-1}$ related to the CO$_2$ in the air [32]; however, as depicted by Shi et al. (2017), it is related to the P-OH vibration, which clearly changes after the solvent extraction reaction [52]. It is observed that the reaction may be carried out by the break of the P=O bond and a further link to the metallic ions, as proposed

in Figure 11. The data obtained show no peak at 3300 cm$^{-1}$ related to structural water, as occurs for Ni and Co by organophosphate [32], which might indicate no water in the organophosphate–REE complexes. No presence of a peak at 3300 cm$^{-1}$ may also indicate this, as well as the absence of NO$_3^-$ [32]. For this reason, the link between Cyanex 923 and La and Y is proposed in Figure 11. Future studies may explore these structures.

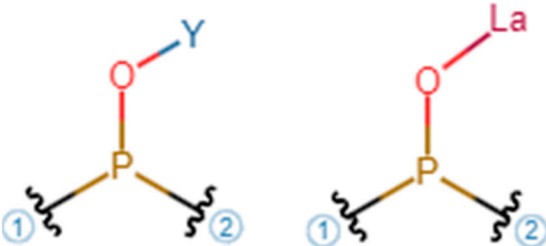

**Figure 11.** Chemical structures between Cyanex 923 and La and Y.

Figure 12 presents the FT-IR spectra of Cyanex 272 20% in kerosene before and after the reaction with the metallic ions. The Alamine 336 did not show a selective separation among the metals studied, so the organic phase was not evaluated. The same peaks were identified, including the P-OH vibration at 2368 cm$^{-1}$, which intensified after the solvent extraction reaction. The peaks located at 1162 cm$^{-1}$ and 960 cm$^{-1}$ are also related to P-O-H vibration. For this reason, it may be inferred that the Cyanex 272 structures with La and Y are similar to Cyanex 923. The data presented in Figure 12 are similar to that reported by Saleh et al. (2002) for La–Cyanex 272 system in toluene [22].

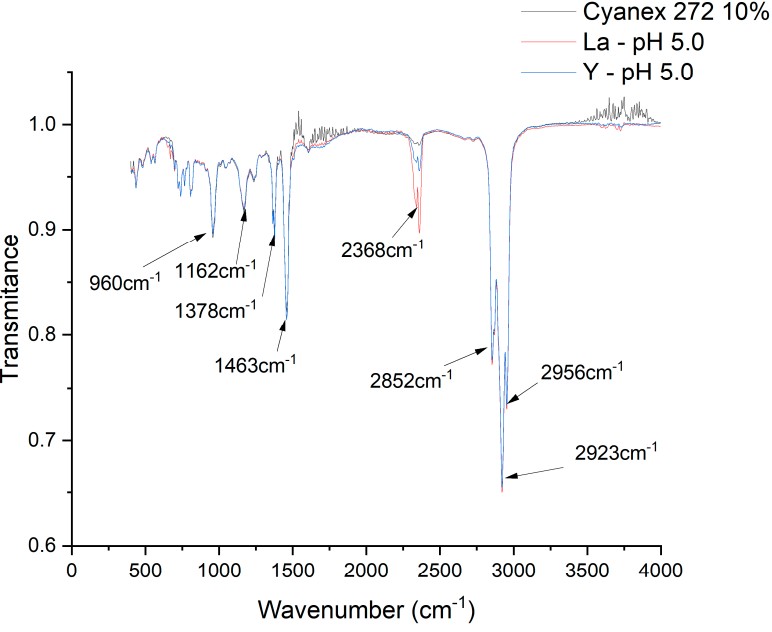

**Figure 12.** FT-IR spectra of Cyanex 272 10% in kerosene, before and after the extraction of La and Y in nitric acid media at pH 5.0.

## 4. Conclusions

The present study aimed to evaluate three different extractants for the separation of La, Y, and Nd. Among the results obtained, it is clear that the extraction order is Alamine 336 >>> Cyanex 272 > Cyanex 923, where both acid extractants were more selective for Y than for La and Nd. The extraction achieved 99% at pH 5.0 in nitric acid media and a Y/La ratio of 2 and Y/Nd ratio of 4 using Cyanex 923. The present study also elucidated the organometallic complexation between Cyanex 923 and Cyanex 272 with Y and La, which may improve separation processes to obtain critical metals from primary and secondary sources.

**Author Contributions:** Conceptualization, methodology, formal analysis, investigation, data curation, writing—original draft preparation, A.B.B.J., N.O.M.d.S., J.A.S.T. and D.C.R.E.; writing—review and editing, A.B.B.J., J.A.S.T. and D.C.R.E. All authors have read and agreed to the published version of the manuscript.

**Funding:** The authors would like to acknowledge the Fundação de Amparo à Pesquisa do Estado de São Paulo and Capes (grants: 2019/11866-5, 2021/14842-0, and 2023/01032-5 São Paulo Research Foundation) for the financial support. This project was developed with the support of SemeAd (FEAUSP), FIA Fundação Instituto de Administração, and Cactvs Instituto de Pagamento S.A. through the granting of assistance to a research project, Bolsa SemeAd PQ Jr (Public Notice 2021.01).

**Institutional Review Board Statement:** Not applicable.

**Informed Consent Statement:** Not applicable.

**Data Availability Statement:** Not applicable.

**Conflicts of Interest:** The authors declare no conflict of interest.

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
