# Peer review of "Structure Investigation of La, Y, and Nd Complexes in Solvent Extraction Process with Liquid Phosphine Oxide, Phosphinic Acid, and Amine Extractants"

_metals, doi:10.3390/met13081434_

Round 1

Reviewer 1 Report

The present investigation focuses on the solvent extraction of La, Nd and Y from synthetic solution by various types of extractants.

Comments

1. Title

The present study focuses on the solvent extraction separation and recovery of La, Nd and Y (not general REE)

Also here, there is not a phosphoric extractant. CYANEX 923 extractant is a liquid phosphine oxide, CYANEX® 272 is a dialkyl phosphinic acid extractant and Alamine 336 is a tri-octyl/decyl amine extractant.

2. and Y/La ratio 2 and Y/Nd ratio 4

Why the Authors have chosen those specific ratios?

3. Three organic extractants were used in this study without purification (purity ≥ 95%). Cyanex 923 (mixture of four trialkylphosphine acids - TRPO), D2EHPA (di-2-ethylhexyl 89 phosphoric acid), TBP (tributylphosphate), and Alamine 336 (tertiary amine).

The above extractants are different from the corresponding mentioning in the Abstract

4. For the experiments, synthetic solutions were prepared using oxides of lantha-92 num (La2O3), neodymium (Nd2O5), and yttrium (Y2O3) dissolved in HNO3

Why HNO3 and what would be the behavior in different type of acid?

Also, we need to know the initial concentrations (g/L) in the solutions

Furthermore, the Authors should have performed trials in higher temperature in order to obtain better extraction and separation factors. Also, in most cases the leaching process usually carried out in higher temperature and the pregnant solution is above 25oC

5. Samples from solvent extraction experiments were analyzed in energy dispersive X-ray 97 fluorescence

Analyses from AAS in aqueous phase are also required for reason of comparison, especially when the concentration are low.

6. 2.2. Materials

2.1 and 2.2 have the same title

7. The extraction of La, Nd, and Y was evaluated for three different types of extractants: chelating phosphinic and phosphoric acids (Cyanex 272)

Why the Authors mention the phosphoric extractants? In this study they use only Cyanex 272, which is a phosphinic one

8. The effect of pH on metal extraction by Cyanex 923 is presented in Figure 5.

So, the separation is not possible. How the Authors explain the pH dependence for La extraction, since the use a neutral extractant?

9. Figure 6 presents the extraction efficiency of Y, La, and Nd by Alamine 336

Low extraction, very low separation

10. Figure 7 depicts the extraction efficiency using Cyanex 272 (C272).

In the Abstract the Authors mentioned that from the results obtained, it is clear that the extraction order is Alamine 336 >>> Cyanex 21 272 > Cyanex 923

According to Figure 7, the extraction is almost complete in pH=5, and Cyanex 272 present better separation index in lower pH. This means that in a counter current solvent extraction system (maybe) the separation in lower pH would be possible

Author Response

(see document attached)

Detailed Response to Reviewers

Manuscript ID: metals-2520068

Title: Structure investigation for rare earth elements complexes in solvent extraction process with phosphine and phosphoric extractants

Many thanks for your valuable comments. We learned greatly from your rigorous attitude towards scientific research. Your comments have an important guide for improving our article. We have studied your comments carefully and have modified the whole manuscript according to your comments. Just to be clear, the manuscript was entirely revised and we believe that all the explanations are now to the point and brief. In our point of view, the revised manuscript is now more acceptable than previously.

Nevertheless, please do not hesitate to send your suggestions and comments to us and we are very glad to receive these because we can always learn a lot from your valuable suggestions and comments. All changes are highlighted in green in the revised manuscript.

Reviewer 1

  1. Title

The present study focuses on the solvent extraction separation and recovery of La, Nd and Y (not general REE)

Also here, there is not a phosphoric extractant. CYANEX 923 extractant is a liquid phosphine oxide, CYANEX® 272 is a dialkyl phosphinic acid extractant and Alamine 336 is a tri-octyl/decyl amine extractant.

Comments: Dear reviewers, thank you for your contributions. Indeed, the title was not in agreement with the entile manuscript. We had changed the title for “Structure investigation for La, Y and Nd complexes in solvent extraction process with liquid phosphine oxide, phosphinic acid and amine extractants”.

  1. and Y/La ratio 2 and Y/Nd ratio 4

Why the Authors have chosen those specific ratios?

Comments: Thanks for your observation. We did not choose these ratios. These data are the result of extraction selectivity between both elements.

  1. Three organic extractants were used in this study without purification (purity ≥ 95%). Cyanex 923 (mixture of four trialkylphosphine acids - TRPO), D2EHPA (di-2-ethylhexyl 89 phosphoric acid), TBP (tributylphosphate), and Alamine 336 (tertiary amine).

The above extractants are different from the corresponding mentioning in the Abstract

Comments: Thanks for your observation. We correct as previous recommended.

  1. For the experiments, synthetic solutions were prepared using oxides of lantha-92 num (La2O3), neodymium (Nd2O5), and yttrium (Y2O3) dissolved in HNO3

Why HNO3 and what would be the behavior in different type of acid?

Also, we need to know the initial concentrations (g/L) in the solutions

Furthermore, the Authors should have performed trials in higher temperature in order to obtain better extraction and separation factors. Also, in most cases the leaching process usually carried out in higher temperature and the pregnant solution is above 25oC

Comments: Dear reviewer, thank you for your contribution to our manuscript. We would like to clarify a few points:

  • Nitric acid was used as the most common for extraction of rare earth elements, as it is used in leaching, as well as hydrochloric, phosphoric and sulfuric acids (see http://dx.doi.org/10.1016/j.hydromet.2016.01.035 / chapter 4.1, https://www.tandfonline.com/doi/full/10.1080/03719553.2016.1181398, http://dx.doi.org/10.1016/j.resconrec.2014.04.007, and http://dx.doi.org/10.1016/j.mineng.2013.10.021 / Fig. 8). The separation of metals may have different results in separation mechanisms. For instance, as depicted in the manuscript in Equations 4-7, the extraction of rare earth elements and in FT-IR data obtained, the extraction occurs without the interaction between REE-NO3 with the organic extractant. However, sulfate or chloride ions may interact with the ions and the extractant, which might result in less extraction efficiencies or no difference.

  Equation 4 [30]

                             Equation 5 [40]

                           Equation 6 [40]

                           Equation 7 [40]

As depicted in our last study - https://doi.org/10.1016/j.seppur.2021.119798:

Despite our last study was related to Sc extraction, the similarities among the rare earth elements may present similar mechanisms. Indeed, the in-depth evaluation would result in another outstanding paper. Thanks for your suggestions for our future studies!

  • The initial concentration of the rare earth elements was 500mg/L as depicted in the chapter 2.1: “The pH was adjusted using HNO3 or NaOH (in beads). The stock solutions (500mg/L) were prepared……”
  • Indeed, the leaching reaction usually occurs at temperatures over 25°C, as 45-90°C previous reported by the authors (see https://linkinghub.elsevier.com/retrieve/pii/S1002072119308919). However, in the case of separation of rare earth elements, due to chemical similarities, the increase in temperature would not be enough to improve the selectivity, as stated by the literature (for instance 10.1016/j.hydromet.2019.04.030). Our study aimed the study of structures formed between La, Y and Nd and the organic extractants.

  1. Samples from solvent extraction experiments were analyzed in energy dispersive X-ray 97 fluorescence

Analyses from AAS in aqueous phase are also required for reason of comparison, especially when the concentration are low.

Comments: Dear reviewer, the samples analyzed were the aqueous phase after extraction reaction. We would like to clarify that the analytical technique energy dispersive X-ray fluorescence spectrometry (EDXRF) is accurate, and it can be used for solutions with metal content higher than 10ppm. Despite the X-ray fluorescence is focused on solid analyses, the technique has been shown efficient for the determination of metals in aqueous media, mainly for concentrations above 10ppm. Tests carried out by the research group have been validated using ICP-OES. Calibration curves were used for the analyzes, which raise the reliability of the results. In addition to this, light elements such as sodium and fluorine are not indicated to be analyzed in X-ray fluorescence. In the present study, the metals analyzed are heavy elements. So, those elements can be analyzed by the technique.

  1. 2.2. Materials

2.1 and 2.2 have the same title

Comments: Thanks for your correction!

  1. The extraction of La, Nd, and Y was evaluated for three different types of extractants: chelating phosphinic and phosphoric acids (Cyanex 272)

Why the Authors mention the phosphoric extractants? In this study they use only Cyanex 272, which is a phosphinic one

Comments: Thanks for your observation we are truly sorry about the mistake. We changed as suggested.

  1. The effect of pH on metal extraction by Cyanex 923 is presented in Figure 5.

So, the separation is not possible. How the Authors explain the pH dependence for La extraction, since the use a neutral extractant?

Comments: The reviewer is correct about his/her observation. Your question makes us to study and revise the literature looking for a question. As a result, we updated the Equation 4 and added an explanation to the process:

“As observed in Figure 5, the extraction of La was influenced by the pH, despite the fact Cyanex 923 is a neutral extractant. It occurs because of nitric acid concentration decreased as the pH increased. As observed in Figure 2, the concentration of HNO3 decreased until zero at pH around 2.8 and a slight increase in nitrate ions formation. From pH 1 to 3, the extraction of La increased 36% to 70%. As depicted in Equation 4, it results in the equilibrium pushed to right for products formation. Also, as the pH in-creases, the H+ in solution compete with the metallic ions for the extractant which was highlighted for La ions [30].”

  1. Figure 6 presents the extraction efficiency of Y, La, and Nd by Alamine 336

Low extraction, very low separation

Comments: Indeed, the extraction of these metallic ions by the anionic extractant was low.

  1. Figure 7 depicts the extraction efficiency using Cyanex 272 (C272).

In the Abstract the Authors mentioned that from the results obtained, it is clear that the extraction order is Alamine 336 >>> Cyanex 21 272 > Cyanex 923

According to Figure 7, the extraction is almost complete in pH=5, and Cyanex 272 present better separation index in lower pH. This means that in a counter current solvent extraction system (maybe) the separation in lower pH would be possible

Comments: We are sorry for the mistake. Actually, it is the opposite, as stated in the revised document.

About the counter-current separation reaction, several changes in separation yield and selectivity as well. It is well stated by the reviewer!

Reviewer 2

I would like to see a discussion by the authors concerning the industrial use of any of the studied extractants  Alamine, Cyanex 272 and Cyanex 923. If I am not wrong P507 is the most widely used extractant in HCl solution, while it seems that the authors focused on nitric solution. Why are they disconnected from industrial practice? Also if are we talking about separating REE from electronic waste,  the authors should give the composition of the pregnant solution.  Usually e-waste, especially magnets, will contain Fe, Pr, Dy and Tb  that should also be separated.  The gaps between La-Nd and Y, make the separation of these elements relatively easy and I am not convinced that the information in the paper is very useful for a practical processing of e-waste. It looks to me that the research work is more academic than practical.

Comments: Dear reviewer, thank you for your contribution and comments. It improved the quality of our manuscript.

As depicted in Xie et al. (2014), Cyanex 923 is also used for industrial proposes for separation of rare earth elements (see http://dx.doi.org/10.1016/j.mineng.2013.10.021). Tunsu et al. (2016) depicts a process for Y recovery from spent fluorescent lamps using Cyanex 923 (see http://dx.doi.org/10.1016/j.seppur.2016.01.048). The same was found for Sc from different sources (see https://doi.org/10.1016/j.mineng.2021.107148 and https://doi.org/10.1016/j.seppur.2021.119798). In the case of P507, it has a similar chemical structure than Cyanex 272, as demonstrated in the figure below (see https://doi.org/10.1016/j.seppur.2017.11.042)

Cyanex 923 has the functional group P=O, while Cyanex 272/P507 has HO-P=O, as depicted in the figure below (see https://doi.org/10.1016/j.seppur.2016.10.039):

As concluded, there is no problem for industrial application of both Cyanex 923 or 272 for separation of rare earth elements, while Alamine 336 is not suitable as its efficiency is lower and no selective was observed.

Regarding the nitric acid solution, the same comment was sent to the Reviewer 1:

Nitric acid was used as the most common for extraction of rare earth elements, as it is used in leaching, as well as hydrochloric, phosphoric and sulfuric acids (see http://dx.doi.org/10.1016/j.hydromet.2016.01.035 / chapter 4.1, https://www.tandfonline.com/doi/full/10.1080/03719553.2016.1181398, http://dx.doi.org/10.1016/j.resconrec.2014.04.007, and http://dx.doi.org/10.1016/j.mineng.2013.10.021 / Fig. 8).

In addition, our study focused on structure analysis of La/Y/Nd and the organic extractants. It is important for potential application for separation of these elements from primary/secondary sources and e-wastes. Such academic study is crucial to bring new highlights for practical applications.

I am not sure that the authors are aware that the selectivity in the separation n of REE should not be measured only at the extraction stage, but that the scrubbing of the loaded organic phase provides an efficient wat to improve the selectivity. The authors should also have discussed the ease of stripping the REE from the loaded organic.

Comments: Thank you for your important contribution.

Table 1 : Please define the recovery rate. Are we talking of synthetic solution of one element? What is the extractant concentration in the organic phase? The low recovery (29%)of Nd with D2EHPA is  difficult to understand. Especially at a pH of 4; unless the organic phase was saturated?  This should have been investigated.

Comments: The extractant concentration in the organic phase was 10%, as depicted in the Methodology section.

For clarification, we did not study the organic extractant DEHPA. It was an error in the materials section.

Line 65 : define Contacts (counter current extraction or co-current extraction??)

Comments: interaction between the organic phase and the aqueous solution.

Equation 1: Define cf, vf….  If f stands for final then it is not an extraction that is calculated, it is a proportion remaining??

Comments: The definition is already in the previous paragraph:

          The amount of ion extracted was calculated by mass balance as depicted in Equation 1, where Ci and Cf are the concentration of the REE before and after the extraction experiment, and Vi and Vf are the volume of the aqueous phase before and after the extraction experiment, respectively. Distribution ratio (D) and separation factor (β) were calculated as depicted in Equation 2 and 3, respectively, where Mt and Ma are the initial and final concentrations of the metallic ions in the aqueous phase.

Also, we updated the Equation 1.

Line 170: What is the purpose of the discussion on Eu if the focus is put on La, Nd and Y??

Comments: Just to clarify to the reviewer, Eu is a light rare earth element, as well as La. As a result, similar insights may be found on solvent extraction process. Also, our discussion related to Figure 3 was important to justify the thermodynamic study previous solvent extraction experiments.

Figure 5: Are we talking of the pH of the pregnant solution or of the equilibrium  pH??

Comments: If the reviewer refer pregnant solution as the synthetic solution used on solvent extraction experiments, it is related to the pH controlled before and during the experiment, as described in the methodology section.

Eq 4 clearly shows that increasing the H+ concentration (Decreasing pH) should not be favorable for the extraction of the REE; is it necessary to experimentally verify the role of the pH?? Knowing the equilibrium constants for the extraction of La, Nd and Y  with the selected extractant should be sufficient to predict the extraction at various pHs.  The same comment applies for Fig. 7.

Comments: We corrected the equation and more discussion was added.

Tables 2 and 3 : It is somehow disturbing to see that the separation factors change with the pH since the pH changes with the amount of extracted metals. Again, are we talking of the pH of the pregnant solution or of the equilibrium  pH??

Comments: The question of the reviewer was answered in the previous comment. Regarding the Tables 2 and 3, it is clearly demonstrated that the increase in the pH declined the selectivity as the extraction of La, Y and Nd increased.

Figure 8 is unreadable.

Comments: We improved the quality of the manuscript.

I don’t think that the section on IR- analysis is very useful?

Comments: We would like to thank the reviewer for all contributions and valuable comments. His/Her comments were important to improve the quality of our manuscript. The FTIR data demonstrated how the structures of La, Y and Nd were formed with the organic extractants. It is poorly reported in the literature mainly for rare earth elements. It demonstrates the novelty of our study.

Reviewer 2 Report

I would like to see a discussion by the authors concerning the industrial use of any of the studied extractants  Alamine, Cyanex 272 and Cyanex 923. If I am not wrong P507 is the most widely used extractant in HCl solution, while it seems that the authors focused on nitric solution. Why are they disconnected from industrial practice? Also if are we talking about separating REE from electronic waste,  the authors should give the composition of the pregnant solution.  Usually e-waste, especially magnets, will contain Fe, Pr, Dy and Tb  that should also be separated.  The gaps between La-Nd and Y, make the separation of these elements relatively easy and I am not convinced that the information in the paper is very useful for a practical processing of e-waste. It looks to me that the research work is more academic than practical.

I am not sure that the authors are aware that the selectivity in the separation n of REE should not be measured only at the extraction stage, but that the scrubbing of the loaded organic phase provides an efficient wat to improve the selectivity. The authors should also have discussed the ease of stripping the REE from the loaded organic.

Table 1 : Please define the recovery rate. Are we talking of synthetic solution of one element? What is the extractant concentration in the organic phase? The low recovery (29%)of Nd with D2EHPA is  difficult to understand. Especially at a pH of 4; unless the organic phase was saturated?  This should have been investigated.

Line 65 : define Contacts (counter current extraction or co-current extraction??)

Equation 1: Define cf, vf….  If f stands for final then it is not an extraction that is calculated, it is a proportion remaining??

Line 170: What is the purpose of the discussion on Eu if the focus is put on La, Nd and Y??

Figure 5: Are we talking of the pH of the pregnant solution or of the equilibrium  pH??

Eq 4 clearly shows that increasing the H+ concentration (Decreasing pH) should not be favorable for the extraction of the REE; is it necessary to experimentally verify the role of the pH?? Knowing the equilibrium constants for the extraction of La, Nd and Y  with the selected extractant should be sufficient to predict the extraction at various pHs.  The same comment applies for Fig. 7.

Tables 2 and 3 : It is somehow disturbing to see that the separation factors change with the pH since the pH changes with the amount of extracted metals. Again, are we talking of the pH of the pregnant solution or of the equilibrium  pH??

Figure 8 is unreadable.

I don’t think that the section on IR- analysis is very useful?

Author Response

(The authors gave the same response as above.)

Round 2

Reviewer 2 Report

The labels for the X  and Y axes are  unreadable.  THis should be corrected.

Author Response

Detailed Response to Reviewers

Manuscript ID: metals-2520068

Title: Structure investigation for rare earth elements complexes in solvent extraction process with phosphine and phosphoric extractants

Dear reviewers and editor Ms. Andrada Simion,

Many thanks for your valuable comments. We learned greatly from your rigorous attitude towards scientific research. Your comments have an important guide for improving our article. We have studied your comments carefully and have modified the whole manuscript according to your comments. Just to be clear, the manuscript was entirely revised and we believe that all the explanations are now to the point and brief. In our point of view, the revised manuscript is now more acceptable than previously.

Nevertheless, please do not hesitate to send your suggestions and comments to us and we are very glad to receive these because we can always learn a lot from your valuable suggestions and comments. All changes are highlighted in green in the revised manuscript.

Reviewer 2

The labels for the X  and Y axes are  unreadable.  THis should be corrected.

Comments: Thank you for your comments. We improved the quality of the figures of thermodynamic simulations.